# Design and Development of Internet of Things-Driven Fault Detection of Indoor Thermal Comfort: HVAC System Problems Case Study

**DOI:** 10.3390/s22051925

**Published:** 2022-03-01

**Authors:** Bukhoree Sahoh, Mallika Kliangkhlao, Nichnan Kittiphattanabawon

**Affiliations:** 1School of Informatics, Walailak University, Tha Sala, Nakhon Si Thammarat 80160, Thailand; knichcha@wu.ac.th; 2Informatics Innovation Center of Excellence (IICE), Walailak University, Tha Sala, Nakhon Si Thammarat 80160, Thailand; 3Department of Computer Engineering, Prince of Songkla University, Had Yai 90112, Thailand; kliangkhlao.m@gmail.com

**Keywords:** machine learning, consciousness prior, heat transfer, indirect measurement, air-handling unit, building sustainability, internet of things

## Abstract

Controlling thermal comfort in the indoor environment demands research because it is fundamental to indicating occupants’ health, wellbeing, and performance in working productivity. A suitable thermal comfort must monitor and balance complex factors from heating, ventilation, air-conditioning systems (HVAC Systems) and outdoor and indoor environments based on advanced technology. It needs engineers and technicians to observe relevant factors on a physical site and to detect problems using their experience to fix them early and prevent them from worsening. However, it is a labor-intensive and time-consuming task, while experts are short on diagnosing and producing proactive plans and actions. This research addresses the limitations by proposing a new Internet of Things (IoT)-driven fault detection system for indoor thermal comfort. We focus on the well-known problem caused by an HVAC system that cannot transfer heat from the indoor to outdoor and needs engineers to diagnose such concerns. The IoT device is developed to observe perceptual information from the physical site as a system input. The prior knowledge from existing research and experts is encoded to help systems detect problems in the manner of human-like intelligence. Three standard categories of machine learning (ML) based on geometry, probability, and logical expression are applied to the system for learning HVAC system problems. The results report that the MLs could improve overall performance based on prior knowledge around 10% compared to perceptual information. Well-designed IoT devices with prior knowledge reduced false positives and false negatives in the predictive process that aids the system to reach satisfactory performance.

## 1. Introduction

A comfortable indoor environment is one of the most critical factors impacting human-life quality (e.g., health, wellbeing, and working productivity performance). Indoor thermal comfort concerns engineering processes to control the environment for satisfying occupants in the building. Heating, ventilation, and air-conditioning systems (HVAC systems) are engineering mechanisms required to handle ambient conditions of indoor thermal comfort to provide occupancy comfort level. Significantly, in the tropical zones (e.g., parts of North America, South America, Africa, Asia, and Australia), outdoor environments are warm throughout the year and influence the indoor environment to become uncomfortable. HVAC systems play a crucial role in controlling indoor ambient conditions by transferring heat airflow from indoor to outdoor. Therefore, HVAC systems, the outdoor environment, and indoor thermal comfort depend on each other to properly control such situations to be comfortable. The case needs engineers to monitor relevant factors from the indoor environment [1], outdoor environment [2], and HVAC system mechanism [3] and to prevent problems that can occur. In practice, engineers and technicians must routinely trace and observe them physically and figure out how unsatisfactory factors can happen. However, enormous systems may run continuously in real-world buildings, and fault detection and diagnosis of indoor thermal comfort based on these complex factors are beyond manual investigation by engineers and technicians. It is challenging to apply advanced technologies to detect and diagnose such relevant factors automatically.

Cardillo et al. [4] reviewed advanced technologies for HVAC systems, e.g., Internet of Things (IoT), machine learning (ML), and radar-based detection. These technologies play an essential role in fault detection and diagnosis of indoor thermal comfort fields [5]. They are critical technologies behind automatic and intelligent systems to address the problem by connecting all relevant things from a dynamic environment. IoT devices percept real-time signals, while ML analyzes them to discover relevant information for supporting engineers and technicians. They can monitor and make better decisions that can prevent situations from becoming worse. Peng et al. [6] proposed an approach for controlling thermal comfort in an indoor environment. Yang et al. [7] proposed a prognostics and predictive approach to detect failure events affecting occupants’ health. Their techniques employed IoT devices and MLs to detect and diagnose failure events caused by HVAC systems failures, and they claimed that such approaches could help decision makers discover problems at an early stage. Xu et al. [8] and Shahinmoghadam et al. [9] proposed an IoT-based system for an indoor environment monitoring to understand thermal comforts. They claimed that IoT-based approaches could uncover vital factors that help technicians detect operating issues according to dynamic environments and repair them on time. Thermal comfort concerns the HVAC system and indoor climate and relates then to conditions of the outdoor environment. Rijal et al. [10] discussed that a good understanding of outdoor environmental factors could help occupants reach a satisfactory level of indoor thermal comfort, while Elnaklah et al. [11] claimed that outdoor environment factors provide vital information for better analyzing the indoor thermal environment for occupant comfort and health. This suggests that fault detection and diagnosis of indoor thermal comfort satisfactions benefit from IoT devices and MLs for discovering information from relevant factors that help engineers explain and interpret the event precisely.

Current research has intensively studied IoT devices and ML-based fault detection and diagnosis in indoor thermal comfort for an occupant’s good living. However, they lack integration of interdisciplinary research between HVAC systems, indoor and outdoor environments. This limitation becomes a problem when occupants cannot acclimate to uncomfortable indoor environments that are not because of HVAC system problems, but because the outdoor environment is worsening, and the heat-transfer system cannot ex-change in time. For example, occupants may experience freezing, mugginess, or sweltering heat, although the HVAC system is turned on. In these scenes, they must call for help from engineers to fix such problems. However, engineers cannot identify and fix the problem immediately and need to observe and collect all relevant factors from physical sites manually (e.g., HVAC system, indoor environment, and outdoor environment) that are time consuming and labor intensive. The system must be turned off during the process where engineers physically check the problem. At the same time, occupants live without an HVAC system and might be more trouble for susceptible occupants such as children, patients, and older people. Therefore, IoT devices and ML-based interdisciplinary research is challenging to address by automatically integrating and analyzing complex factors that detect and diagnose problems early.

This research encodes all relevant factors concerning interdisciplinary study, HVAC systems, the outdoor environment, and the indoor environment. It aims to describe what factors may co-occur with failure events that cause occupants to feel uncomfortable. Our research consists of two parts: (1) hardware designs and developments to connect relevant observations from environments and (2) software designs and developments to detect and diagnose events of interest. These can help engineers insightfully understand the problem in advance to plan strategies to minimize the crucial situation proactively. The main contributions of this research are:Explorations of random variables to represent relevant knowledge from interdisciplinary factors;Physical designs of an IoT device to collect real-time factors of events as states of random variables;Proof of correlations between factors to explain indoor thermal comfort;Proposal of machine learning models enhanced by interdisciplinary factors.

This research is structured as follows: Section 2 reviews background knowledge and related works; Section 3 proposes explorations of random variables to represent relevant knowledge from interdisciplinary; Section 4 contributes an IoT device to collect real-time observations as states of random variables; Section 5 analyzes correlations between interdisciplinary factors to explain thermal comfort; Section 6 proposes machine learning models enhanced by interdisciplinary aspects; Section 7 summarizes the design and deployment of IoT-based measurement for thermal comfort in the indoor environment.

## 2. Related Works

This section proposes background knowledge for understanding relevant factors whose effects may impact indoor thermal comforts. Recent advances in the HVAC system, outdoor environment, and indoor environment are investigated to show current progress and future trends.

### 2.1. Thermal Comfort and Life Quality

Uncomfortable environments may impact occupants’ quality of life, which is relevant to work productivity and health. In health care sectors, controlling indoor thermal comfort plays a vital role that can help physicians understand patients’ problems that they can diagnose and thus plan for proper treatment. Aghamohammadi et al. [12] examined that toxic environments caused by thermal comfort are highly associated with somatic symptoms (e.g., insomnia, headache, fatigue, and dizziness). They claimed that the statistical significance between them shows strong correlations. Sun et al. [13] discovered that bad thermal comfort has significantly co-occurred with children’s respiratory diseases (e.g., asthma, dry cough, and pneumonia). Tsang et al. [14] and Cao et al. [15] discussed that sleep quality is a factor of life quality, and poor thermal comfort directly affects it. They summarized that reasonable control of indoor thermal comfort could help occupants reach good life quality. Such reasonable control can only be accomplished if the engineers understand outdoor environments and HVAC system factors.

Ma et al. [16] discussed measuring the right factors relevant to occupant wellbeing based on thermal comfort. They reviewed recent trends and found that understanding thermal comfort depends on how related random variables are well defined. They must be designed to cover all conditions in environments because computer systems can monitor and analyze the situation from the environment based on random variables. They discussed that designing and developing random variables aligned with IoT and ML technologies is challenging to sense, detect, and diagnose such factors. The following section will review how ML is influential to observe and identify such relevant factors.

### 2.2. Fault Detection

A fault detection system is a computational process to determine abnormal events, failures, and malfunctions, which can be unsatisfied to people and damage properties [17]. A fundamental principle of fault detection requires knowledge to reason the possible abnormal events. Fault detection needs human-like perception to sense what happens in a particular environment [18]. Such input is a vital feature essential to detecting the fault. Sensor technologies such IoT devices play as perceptions to input signals into the systems.

Automatic fault detection may employ a rule-based system (if-part analysis) determined by domain experts. However, in indoor thermal comfort, rule-based systems may not cover in case of complex events that perform malfunctions. For example, an uncountable indoor environment may correlate to an outdoor setting and HVAC systems that can randomly occur without clear rules to identify. Shi et al. [19] researched automated fault detection and diagnostics for building systems. They discussed that ML is a core technology that can help systems detect a fault in indoor thermal comfort. Chegari et al. [20] reviewed recent trends and future directions for fault detection systems in indoor thermal comfort. They showed that ML is highly capable of dealing with failures in complex environments. The following section will show how ML may help the fault detection systems.

### 2.3. Machine Learning (ML)

ML is the brain of software agents to acquit knowledge and make intelligent decisions [20]. It allows software agents to imitate human-like intelligence to solve particular problems or tasks such as fault detections. It learns such problem solving using experience provided by engineers and practitioners and improves learning performance according to varieties of affairs [21]. In this way, indoor thermal comfort fault detection must employ ML to encode engineer and practitioner logic on how to deal with the problem to alleviate time-consuming and labor-intensive tasks [22].

Experiences are directions that allow ML to encode reasonable logic, and engineers and practitioners may guide ML on identifying the faults by labeling sample data. It is called supervised learning, which helps software agents fit the functions between sample data and its label. Finally, software agents can automatically distinguish and classify which events are regular or faults that help occupants make decisions.

Standard supervised ML algorithms can be divided into three approaches: geometry, probability, and logical expression, as studied by Flach [23]. A Decision Tree (DT) is an algorithm-based if-part expression divided instance using a Boolean valued function (if ***X*** then-part ***Y***). Artificial Neural Network (ANN), k-Nearest Neighbors (k-NN), and Support Vector Machine (SVM) are distance-based metrics to compute dimensions in space (***X***- and ***Y***-axis). Naïve Bayes is a probabilistic algorithm to calculate features using a conditional probability distribution P (***Y***|***X***). The three approaches employ different algebraic functions but can compute and classify outcomes. However, the critical facts are that the outstanding performance of ML algorithms depends on the quality of experiences (e.g., nonsense in, nonsense out). Therefore, the design and development of input technologies such as IoT devices must help ML algorithms achieve good performance. Our research question is “if we design and develop the input process well, then ML algorithms can better perform the task of indoor thermal comfort better”. In the next section, we will explore knowledge contributed to the research question.

### 2.4. Related Works Based on IoT and ML

This section is dedicated to examining current research based on IoT devices and ML for controlling indoor thermal comfort. We divide our reviews into three aspects: indoor environments, outdoor environments, and HVAC systems.

Good indoor thermal comfort relies on a well-understood outdoor environment that helps engineers operate heat transfer, and it is currently an impressive research field. Kükrer and Eskin [24] researched how the outdoor environment in different building zones affects indoor thermal comfort. They summarized that the outdoor climate directly affects indoor thermal comfort, and its factors should be employed to control occupants’ satisfaction. Chegari et al. [25] found that outdoor environmental factors and indoor thermal comforts are highly associated. They analyzed outdoor environments related to building design, such as opaque and glazed walls, shading devices, and thermal ventilation. They highly influence indoor thermal comfort, and building designers must consider them to provide occupants’ satisfactory feelings. In conclusion, these outdoor environment factors influence the indoor thermal comfort where 90% of occupants spend their time in the building. Moreover, occupants’ wellbeing and health rely on this concern, and the indoor thermal comfort needs more contribution to help people adapt themselves based on a dynamic outdoor environment [26].

The factors of discovery of indoor environment thermal comfort is recently an important issue and has been widely paid attention to from various research aspects. Song et al. [27] reviewed data-driven research of human comfort in the indoor environment. They found that IoT devices and MLs are the technologies that were most applied to discover indoor environmental factors. Zhang et al. [28] studied the roles of temperature and humidity that impacted thermal comfort in the indoor environment, while Kong et al. [29] studied how temperature and humidity correlate to indoor thermal comfort. They summarized that indoor temperature and humidity are dynamic, and controlling them using IoT devices and MLs to be stable is challenging because they are random, unknowable patterns, and automatic systems to handle them are demanded.

Automatic HVAC systems are a mechanical control of fresh airflow movement in a building. They offer intelligent mechanisms to balance indoor and outdoor temperature and humidity. IoT devices and MLs are the heart of automatic HVAC systems that mimic human-like manners to perceive and detect indoor thermal comfort events and control them to satisfy occupants. Yan [30] proposed an automatic approach for monitoring a mechanism of HVAC systems that can identify instrument malfunction early. He applied IoT devices for reading signals from HVAC systems, employed ML to detect fault mechanisms, and discussed that IoT devices and MLs are potent tools for automatically controlling HVAC systems. Li et al. [31] proposed a system for detecting fault mechanisms in HVAC systems. They employed sensor-driven information to train machine learning models that could identify abnormal events effectively.

Although the work has been intensively researched in various fields (e.g., outdoor environment, indoor environment, and HVAC systems), no work explains the factors of complex problems that may co-occur or co-cause occupant discomfort. Following simple questions, “*Why are we suffering from muggy feelings although the HVAC system has been running with perfect mechanisms?*” and “*Why do we feel cooler or hotter than what we set it on the thermostat?*”, it is hard to answer such occupant questions immediately, primarily when HVAC systems produce the problem that originates uncomfortability in an indoor environment. There are recently no standard measurements to diagnose problems without engineers and technicians who must manually check HVAC systems on the physical site. They might break up the HVAC systems to set up sensors to collect relevant information automatically. However, it is a high cost and requires advanced knowledge in heat transfer-based technologies to do so.

These need new contributions to study co-effects from interdisciplinary fields to explain how these events have happened that can be employed to predict such problems automatically. Therefore, our research aims to fill this gap by contributing IoT-based measurements for indoor thermal comfort caused by HVAC system problems and dynamic outdoor environments using machine learning for automatic detection events.

## 3. Overview System of Thermal Comfort Diagnosis

This section proposes the overview architecture to connect indoor and outdoor environments and HVAC systems. These can help engineers realize and deal with the problem by using real-time digital information to monitor and detect them early. The overview architecture is shown in Figure 1.

Figure 1 consists of five components: (1) home connecting, (2) data sensing, (3) data preprocessing, (4) comfort interpreting, and (5) application. Home connecting represents environments where the system needs to observe indoor thermal comfort and its outdoor conditions. It is a sensory component acting as engineer perceptions to observe indoor and outdoor situations, and the HVAC system functions to figure out physical sites. Home connecting requires installing sensor-related devices and software to activate the system running when devices are turned on. Such software also controls devices to frequently read analog signals from the environment and connects physical sites to a cloud-based platform to stream analog signals to the data sensing component. Data sensing is the first component of a cloud-based platform site. It determines signal features from each sensor and stores them into digital formats. Data sensing holds only raw digital data from physical sites, but it cannot be employed for thermal comfort detection and diagnosis and must be prepared first. Data preprocessing transforms raw digital data into a machine-readable format. It discretizes raw signals to thermal comfort events and describes their semantics using descriptive statistics such as mean, standard deviation, min, and max. These statistical factors are employed as input features for machine learning to model thermal comfort knowledge. Comfort diagnosis is the system’s heart that produces outcomes using a correlation-based machine learning model between indoor satisfactory feeling factors and the HVAC system’s outdoor environment. The predicted outcomes are visualized to engineers, technicians, and occupants that help them know what is happening and how to adapt themselves at that moment to fit with a different environment.

A sensor-related device is a hardware component that needs to be designed and developed before running the system. In the next section, we dedicate to research on how to perform such a device covering all the system’s environmental factors.

## 4. Random Variables (RVs) for Environmental Representation

Events of environments and HVAC systems are fixed (e.g., a state is known), but they occur randomly (e.g., a time is unknown). For example, Thailand’s outdoor temperature is fixed with specific values between 22.1 and 35.4 °C, but it is unknown to which value points will happen. It can be understood how nature works, and we need to model such understanding into a machine-readable format. We need measurable functions to encode these possible values to the actual numbers. RVs encode characteristics of the random phenomenon when the characteristics are fixed and defined as events. However, the events are unknown to when and where they will occur. RV is a mathematical object and allows software agents to compute its events (i.g., estimated, calculated, and classified). We employed RVs to model relevant factors based on the HVAC system, outdoor environment, and indoor environment, which is listed in Table 1.

Table 1 details low-level information using continuous RVs and their possible values. The low-level information is directly sensed from physical sensors without semantic meaning that humans can understand. RVs are determined state ranges according to their natures. For example, temperatures are set to 0–60 °C because 56.7 °C is the highest temperature recorded on Earth. People cannot deal with temperatures high than this, which causes death from hyperthermia.

Electric current is a new assumption to model HVAC load behavior that ranges between 0.00–20.00 amperes. It is proposed to measure the effectiveness of an HVAC system. The assumption is “*If we observe*
**HVAC airflow**
*in different conditions, the characteristics of power consumption behavior must be dissimilar from each other”*. Because the hardest airflow is transferred, more power is needed to drive the HVAC system. We can employ electric current as an indirect measurement of airflow patterns in indoor thermal comfort. In other words, it aims to simplify the complex testing of mechanical heat transfer and address the labor-intensive and time-consuming tasks for engineers to spot problems on physical sites replaced by IoT technologies to detect instead.

Timestamping indexes a transaction of all relevant factors that helps software agents identify thermal comfort events. It represents electronic timestamps (year, month, day, hour, minute, second) that are crucial for recording when such an event occurs. The electronic timestamp is recoded by a computer system (e.g., microcontroller unit) based on the Time Stamp Authority (TSA) server standard. Software agents can exchange information referring to the same event time indexing.

In thermal comfort, we can understand that all factors from physical sensors are continuous, but we need high-level information to make immediate decisions. For instance, we do not consider the temperature of 32.4854527… °C with ±0.3 °C but need to know whether it is hot (higher than 27 °C), it is cool (lower than 20 °C), or it is acceptable (between 20 and 27 °C). The semantics of data points is essential information, and low-level information models semantics of humans, understanding how they feel after sensing signals from environments. In simple words, low-level information is derived from low-level information with human understanding representations.

High-level information models relate factors according to interdisciplinary views (e.g., HVAC system, indoor environment, and outdoor environment). The low-level information from Table 2 models signals from physical sensors working as direct human perception. The high-level information model semantic, understanding how a human feels, needs prior knowledge to infer such feeling. Therefore, the low-level information is raw materials for converting into high-level information. For example, to encode indoor feeling, indoor temperature and humidity are primary sources of whether they are balanced to comfort humans or needed for control and adjustment. Our high-level information for environmental measurement is represented in Table 2.

Table 2 shows high-level information using continuous and discrete RVs to encode relevant states as possible events in the human-level understanding called prior knowledge. **Outdoor Feeling** and **Indoor Feeling** are synthesized based on the balance of two primary sources: temperature (**Temp**) and relative humidity (**RH**). Prior knowledge for balancing between temp and RH was introduced by Xiong and Yao [32], of which we can program this prior knowledge and then transfer it to the microcontroller, which allows agents to transform the raw signal to human understanding. For example, if the temperature is between 26 and 27 °C, an expected relative humidity should be between 40% and 50% that is an optimal dew point for comfortable occupant feeling.

Humidity and temperature differential (Hum–Temp Diff) is the difference in temperature and humidity between indoor and outdoor environments. Hum–Temp Diff implies the heat transfer process of **HVAC Airflow,** how it can move heat from inside to outside, and how it can balance indoor and outdoor humidity. Engineers and technicians utilize them to diagnose problems in thermal conform. In this way, we need prior knowledge to transform raw data into Hum–Temp Diff. Prior knowledge of this concern was initially carried out by Humphreys [33] and Zhang [34], who studied relationships between indoor and outdoor environments. We model Hum–Temp Diff based on three aspects: **In****-Out Temp Diff**, **In****-Out Humi Diff**, and **Thermostat Diff**. **In****-Out Temp Diff** and **In****-Out Humi Diff** represent the differential between indoor and outdoor using the simple mathematic minus operation of two primary sources. **Thermostat Diff** models the differential between exact indoor temperature (sensor measuring) and temperature expectation (temperature on HVAC thermostat set up by the user).

The **HVAC Airflow** reflexes the HVAC system performance in the process of dew point balancing. We designed **HVAC Airflow** RV based on the suggestion from engineers who are experienced with problem solving in indoor thermal comfort. They claimed that most indoor thermal comfort problems relate to HVAC system performance. In contrast, figuring out such a problem is complex and requires high heat transfer mechanical engineering skills. For example, **HVAC Airflow** is good, bad, or worst, and engineers must respond differently to such events to fix or prevent uncomfortable situations.

**Time of Day** roughly models a period of occurring events, and people do not recognize it with complete information. For example, they do not say, “*it was very muggy on Monday 2nd, August 2021 at 22:57:36”* instead, they may say, “*it was very muggy last night*”. They show that timestamping can be discretized when they want to explain thermal comfort events. We then modeled the **Time of Day** by discretizing its states into the morning, afternoon, evening, and night from the date–time factor that software agents can understand in the same manner with human-like understanding.

The following section will introduce a new design and deployment of IoT devices for thermal comfort application aligned with random variables for environmental representation.

## 5. IoT-Based Environment Perception

The design and development of sensor-related devices are challenging because they need to percept multiple factors from different aspects. To develop sensory components, one must understand how nature works based on interdisciplinary research between HVAC systems, indoor environment, and outdoor environment. This section aims to elaborate on how agents perceive relevant factors using computing and IoT technologies.

### 5.1. Environmental Factor Observation

The hardest part in thermal comfort is identifying critical environmental factors and representing them in a machine-readable format. We aim to design sensor-related devices using fewer sensor models but can represent environmental factors effectively, because we consider opportunities to expand this device prototype to be a product for future businesses that should be applied in practice with reasonable price that occupants can afford on them in real-world.

We begin our design by considering the indoor environment that concerns relative humidity and temperature. According to the literature review, feel-like comfort depends upon the balance between humidity and temperature, which recent research has proven. We decided to employ relative humidity and temperature to represent our indoor environment because the HVAC system balances relative indoor humidity and temperature by transferring indoor airflow to an outdoor environment that wishes to obtain the best indoor thermal comfort.

The effectiveness of an HVAC system relies on how airflow exchanges between indoor and outdoor environments. Unfortunately, measuring such airflow requires a high skill of mechanical engineering and needs to customize the physical model of the HVAC system to install sensors and trace the airflow quality. Moreover, modification of the physical model will automatically expire the warranty, which is a big problem, and occupants are concerned about this; thus, we should avoid this solution. We then designed an indirect measurement of HVAC system airflow quality. It imitates when physicians want to treat their patients by initially measuring the symptom to diagnose diseases. For instance, patients’ runny noses may be diagnosed with flu, or skin rashes may be diagnosed as allergies. Observing symptoms is basic information and might not be complete to confirm diseases but is still helpful to warning physicians early to treat first aid and prevent patients from becoming worse. We employ this intelligent manner to diagnose the HVAC system problem through its electricity consumption. In simple words, the highest level of electricity consultations represents the symptom of the most demanding mechanical engine of HVAC system works because of airflow problem.

The outdoor environment is another factor that influences indoor thermal comfort, which depends on its variation. For example, outdoor temperature and relative humidity vary according to location, season, and duration, which are out of control. We encode the variation by measuring outdoor temperature and relative humidity.

We aim to design sensor-related devices by covering three relevant fields: indoor environment, outdoor environment, and HVAC system, and need to specify physical sensor models, their factors, possible ranges, and error rates. We choose the sensor models that can function on relevant fields, and the details of these needs are shown in Table 3.

Table 3 shows that physical sensor models used in indoor and outdoor environments are different, although their factors are similar. Sensor models for the outdoor environment need waterproof factions and endure bad situations such as heavy rain and intensive sunlight. The physical sensor model can be replaced by alternate model brands available in the market zone. For instance, if AM2315 I2C Single Bus might not be available for the outdoor environment, it can be replaced by LHT65 LoRaWAN or something else as long as their factor properties are similar. However, these sensors are analog-read, and hardware can only be run when the controller commands them.

The roles of the controller are to control sensor streaming signals from the environment and connect them with a cloud-based platform. It is an essential part of our system. Moreover, the controller is a source of timestamping to produce the index of events. It connects to a timestamping server using computer software that needs to be programmed. We apply the ESP32 microcontroller unit, multicore processors for three main reasons: (1) low cost of usage and purchase, (2) low-power consumption, and (3) built-in Wi-Fi and Bluetooth microchip connection with open source. Ioannou et al. [35] and Karthick et al. [36] employed ESP32 in a real-time system for controlling sensors and transferring their signals to a cloud-based platform. They discussed that such a microcontroller could process information effectively. Therefore, we selected ESP32 as a microcontroller to control relevant sensors and connect them with our cloud platform. Microcontrollers can be used differently according to the purpose and available items. For example, the popular microcontrollers applied in air quality monitoring systems are ESP8266, Arduino Uno, and Raspberry Pi [26].

The following section details the design and development of the device perception to connect the sensor models.

### 5.2. Design and Development of Device Perception

Sensor-related devices must sense all relevant factors in a parallel and real-time manner. They need a specifically well-designed and functional prototype to fulfill such a manner. We propose utilizing sensors detailed in Table 3 and interconnecting them using an ESP32 microcontroller, a medium between real-world environment and cloud platform. ESP32 connection employs Message Queue Telemetry Transport (MQTT) [37], the standard messaging protocol for the Internet of Things (IoT), which depends upon communication network bandwidth and can undesirably be down. We address this concern by plugging the secure digital card (SD card) to store continuous signals if the system fails and resend it to the cloud again when the connection is recovered. Our blueprint of circuit design and development and prototype is shown in Figure 2.

Figure 2 shows the design and implementation of a final prototype that can sense the thermal factors from the environment. Figure 2a,b shows the creation of the PCB sensor connection in the device prototype. It consists of five elements. DHT22 is selected to observe the temperature and relative humidity as indoor conditions and outdoor conditions. SCT-013 is for HVAC systems’ electrical usage and SD Card reader for temporal data storage before streaming to cloud storage. Lastly, we control relevant sensors based on the ESP32 microcontroller to convert analog into digital signals using the software. The final prototype we used is shown in Figure 2c. The best of our prototype is compact and able to plug and play the sensors in case the sensors are defective, and this provides an easy way to replace the new one.

For the software aspect, we implemented the software for sensor controlling using MicroPython [38]. Micropython is a low-level Python operating with advanced features to transfer commands from general Python to embedded systems. Based on Micropython, they are open source and ready-used libraries to control physical sensors such as DHT, AM2320, and EmonLib for monitoring electric current consumed, which works well with SCT-013 [39]. We streamed signals from all factors to Google Cloud Internet of Things (IoT) Core [40], data collection in the cloud, ready to analyze and visualize to support engineers interpreting thermal comfort situations.

This prototype will be installed to observe environments as a human-like perception-based sensory device. Our design and development process was consulted and guided by mechanical engineers in heat transfer and thermal comfort. We aimed to minimize relevant factor sensing, which is crucial for future commercialization but is still good enough to help them understand thermal comfort.

The following section will employ this signal to analyze how such factors relate to indoor thermal comfort and evaluate relationships between system load behaviors and airflow patterns. RVs from Table 3 will show how they can affect indoor thermal comfort.

## 6. Indoor Thermal Comfort Measurement Case Study

This section highlights how to measure relevant factors based on RVs. Both the outdoor environment and HVAC system can correlate the indoor thermal comfort factors. Our research questions are (1) “*Do the factors sensing from the indoor environment, outdoor environment, and HVAC system have a statistically significant relationship between them?*” and (2) “*Is there a statistical association between indoor environments, airflow patterns, and power consumptions?*”. Herein, we do not ask about outdoor temperature and relative humidity that may impact indoor thermal comfort. They are well-known factors that most current research showed highly impact indoor thermal comfort [41,42].

### 6.1. Design of Experiment: Environmental Setup

Our experiment was set up in the living room at the computer engineering department, Prince of Songkhla University, Songkhla, Thailand. The room size is 6 × 12 m and consists of six LED light panels, one LED television, and one personal computer. The airflow model of the HVAC system uses air condition based on an on-off system with 30,000 BTU (British Thermal Unit), and its maximum power consumption is 11 Amp per hour reported on the paper plate model. The thermostat of the air condition system was set up to 25 °C, and the air velocity was fixed to an auto mode.

To maintain the conditions of the outdoor environment, we set the experiment date from January 2020 to February 2021. Songkhla has a tropical climate, which is hot all year round (30–34 °C).

This indoor environment was fixed (e.g., no additional object is moved in or moved out of the room) during data collecting to ensure no radiant heat from the additional object, which can be biased to indoor thermal comfort during the experiment. The daily occupancy pattern in the room was regular and consistent across the days. Three researchers lived daily in the room from 9.00 a.m. until 5.00 p.m. The normal activities were meeting and performing research. According to the hot weather, people wore hot weather clothing that had a small effect on the heat exchange between the skin/clothing boundaries. The room and its facilities required for our experiment are shown in Figure 3.

Figure 3 shows that our prototypes were installed in two positions where Prototype 1 is for the indoor environment and Prototype 2 is for the outdoor environment and HVAC system. The prototype was programmed to control all sensors by collecting data every two seconds. Prototype 1 measures indoor temperature and humidity where its position must be set close to air condition. It can represent how the air conditioner’s thermostat works, which is helpful for engineers to diagnose indoor thermal comfort problems caused by the HVAC system. Prototype 2 measures two situations: (1) outdoor temperature and humidity and (2) electricity consumption, which consider only the HVAC system. Its position must be installed on the outdoor building site (e.g., not on partitions between buildings) to directly perceive sunlight, wind, shade, and rain. These can model temperature and humidity random variables for each condition. For instance, intensive sunlight may relate to high temperatures, and heavy rain may relate to high humidity. Electricity consumption is measured through current lines (e.g., an average of three phases) that are supplied to the HVAC system. The current measurement is wired to a compressor directed to the outdoor prototype.

Our prototypes will read signals from indoor, outdoor, and HVAC systems, store them in the SD card, and stream them to the cloud-based platform. In the next section, we will elaborate on how such signals can expose the characteristics of each situation.

### 6.2. Normal Distribution of Physical Sensor-Based Relevant Factors

This research aims to employ basic sensors and measure how HVAC systems can affect indoor thermal comforts but can uncover such factors in the standard of professional equipment. Normal distributions of each relevant factor are essential information that engineers must employ to identify problems for indoor thermal comforts. It exposes behaviors of factor characteristics based on descriptive statistics.

Temperature and relative humidity of outdoor and indoor environments are well-known factors to measure thermal comfort that are statically changing together in nature [43]. Power consumption of HVAC systems is another factor that plays a crucial role in building and energy fields. Recently, engineers and technicians have manually measured how the HVAC systems consume energy during the process with professional equipment. However, problems caused by HVAC systems that affect indoor thermal comforts through energy consumption behavior are not considered. Moreover, it is not assessable to ordinary users to measure how they can be related. We collect these signals from our prototypes using a randomized controlled trial (RTC), design trial intervention based on three categories based on HVAC system behaviors: (1) good airflow, (2) bad airflow, and (3) worst airflow. The HVAC system behaviors have been categorized and determined by engineers and technicians who have respondent to maintain the HVAC system. These categories provide an insightful information for respondents to diagnose the causes of system behavior and the urgence of the responding.

In this case study, we control the air conditioner’s filters by blocking airflow streaming through the evaporator coil as if filters lack proper maintenance that cause them to be dirty. This situation is a typical mistake that occupants do not realize and are concerned about, while it is a primary cause of problems in indoor thermal comforts.

We set 0% (e.g., filters are completely blocked, and no air can flow through the system), 20% (e.g., filters are blocked in 80% of their areas, which means less air can flow through the system), and 100% (e.g., filters are spotless, and air can provide perfect flow through the system) of airflows. At the same time, other conditions such as the amount of refrigerant, thermostat sensor, condensate drain, compressor, and fan controls were pre-checked and proved that the HVAC system could run perfectly. The RTC is a powerful method to simulate rare events, theory of black swan events, such as 0% and 20% airflows that are difficult to happen but are likely possible in real-world conditions. We did not measure the other condition because we need to approve our research questions with basic experiments. Nevertheless, it makes sense to confirm that HVAC systems directly impact indoor thermal comforts, and indirect measurement based on cheap sensors can initially identify HVAC systems problems.

We sample the data for three scenarios from signal collections where the outdoor environments are fixed by collecting in the same season and employing a standard of descriptive statistics to uncover a normal distribution of each relevant factor. Samples for each scenario are categorized based on airflow conditions: good, bad, and worse filters. We used sample sizes of 1600 transactions to statistically analyze each airflow conditions. Good airflow behavior was collected from 9.00 a.m. until 5.00 p.m. on Wednesday, 21 July 2021. The summarization of good airflow is shown in Table 4.

Table 4 clarifies the distributions of each random variable that the mean of indoor temperature (**I****-Temp**) is 25.45 ± 0.26 °C, converged to the ideal temperature setting on the thermostat, which is 25.00 °C. The indoor humidity (**I****-RH**) is 46.24 ± 2.07%RH that is perfectly comfortable for occupants according to the standard of the dew point prior knowledge (see the detail in [32]). The electric current (**Ampere**) consumed 5.32 ± 3.35 A that is around half the default value set on the paper plate model that is 11.00 A. It is the power used while transferring the heat from indoors into outdoors. The outdoor environment is the worst condition where the outdoor temperature (**O****-Temp**) is 31.62 ± 0.64 °C, and the outdoor humidity (**O****-RH**) is 80.18 ± 5.20% RH. Engineers interpret that the performance of the HVAC system is a good condition that can control indoor thermal comfort well, although the outdoor condition is worst.

We sampled the data of bad airflow from 9.00 a.m. until 5.00 p.m. on Friday, 2 July 2021. The statistical description is shown in Table 5.

Table 5 shows the distributions where airflow condition is bad and the average trend of outdoor temperature and relative humidity is equally likely with Table 3 (when considering together their standard deviations). The indoor environment, temperature, and relative humidity are entirely different from Table 3, and the indoor thermal comfort is uncomfortable according to the standard dew point. The temperature of 22.98 °C with 40.83% of humidity is dry for occupants to spend their time working. Significantly, the first quartile of indoor environment shows where the temperature is less than 23.00 °C, and humidity is less than 40% RH. They are undefinable, or humans might not be able to spend their lives in such the worst situation because it causes them to be sick. It confirms that the bad condition of the HVAC system influences indoor thermal comfort where the outdoor environment is similar to the good condition of the HVAC system from Table 4. The exciting point is electricity consumption, where the average power of bad airflow is used 6.1 A, and good airflow is used 5.32 A, in which their behaviors are different. Moreover, their first quartile shows that bad airflow consumes 6.47 A while good airflow consumes 0.18 A, highlighting the patterns of different behaviors between the two scenarios.

The worst airflow condition was set up from 9.00 a.m. until 5.00 p.m. on Thursday, 8 July 2021. The statistical description is shown in Table 6.

Table 6 shows that the distributions of indoor environments are comfortable where the average temperature is 26.15 °C and the average relative humidity is 38.09% RH. The rest of the average factors are similar to the bad conditions in Table 5. However, when we consider the details, we found that indoor thermal comfort is more dynamic than other conditions. The first quartile of the indoor environment is dry, where the average temperature is 23.90 °C. The average relative humidity is 34.10% RH. The median is quite acceptable, where the average temperature is 26.50 °C, and the average relative humidity is 40.10% RH. However, it is not comforting, as the thermostat is set up while the third quartile is warmer than the rest. It shows that indoor thermal comfort in a day is changed entirely from cooler to warmer, which can harm occupants both for health and productivity. Susceptible occupants such as older adults and patients should avoid this situation to prevent the worst effect. The behavior of electricity consumption is similar to bad conditions but still has different features, such as that the median and the standard deviation are higher and narrower than the rest.

In conclusion, the behavior of the HVAC system directly associates with indoor thermal comfort, and controlling thermal comfort must interpret how airflow works. Moreover, power consumption highly correlates to airflow behavior, and we can say that power consumption is an indirect measurement of HVAC systems. It confirms that our IoT-based design and development prototype can percept key signals from environments, and we can utilize this to understand indoor thermal comforts.

### 6.3. Correlations between Continuous RVs

The goal of this section is to answer the research question “Do the factors sensed from the indoor environment, HVAC system, and outdoor environment have a statistically significant relationship between them?”. It aims to measure correlation coefficients between continuous variable factors that can be employed to estimate how relevant factors can strongly co-occur if one of them is observed.

We applied Pearson’s correlation (Pearson’s R) to perform a measurement that can return the statistical relationship in three possible ranges of outcomes. Pearson’s R is the symmetric relationship between two variables (**Ampere** co-related to **I****-Temp** equals **I****-Temp** co-related to **Ampere**), regardless of the types of their factor unit (e.g., °C or % RH). Ranges of Pearson’s outcomes are between +1 to −1, where the positive score (+1) defines if a relationship of one random variable factor increases, then the magnitude of other random variable factors will increase. The negative score (−1) represents a relationship of one random variable factor increasing, and the magnitude of the other random variable factors decreasing. Neutral (0) indicates that changing the relationship of one random variable factor is not associated with the rest. The behavior of events co-occurring from Pearson’s R helps us make sure that our design of perception device helps support engineers and technicians to identify the problem of thermal comforts. The correlation coefficient between continuous random variables is shown in Figure 4.

Figure 4 shows correlation coefficients between factors from the indoor, outdoor, and HVAC systems based on the M*_i_* × M*_j_* matrix. Each cell represents symmetric correlation (R*_ij_*) measuring the linear association between row *i*th factor and column *j*th factor. The color gradation from blue to white represents a positive relationship (+1 to 0), and the color gradation from white to red represents a negative relationship (0 to −1).

We can interpret the degree of correlations according to the cell; the pairs of **O****-Temp** and **O****-RH** and **O****-Temp** and **I****-RH** are perfectly opposing. In comparison, the pairs of **I****-Temp** and **I****-RH** and **I****-Temp** and **I****-RH** are perfectly positive regardless of airflow conditions because these correlations are changing together in nature. Their values are slightly different in each airflow condition but still obverse the features and directions of such correlations. Especially, **Ampere** in all conditions is relevant to the remaining factors (overall value lies over ±0.50) that can interpret whether it can be used to observe how the rest of factors appear if **Ampere** changes its correlations or directions.

When we compare the worst condition with the rest, some cells change the correlations’ directions entirely. For example, the pairs of **Ampere** and **I****-****Temp**, **I****-****Temp** and **O****-****RH**, and **O****-****Temp** and **I****-****Temp** change the directions (from blue to red or vice versa) that means airflow plays a critical role in indoor thermal comfort.

In conclusion, most of them are likely co-occurring, and they can be used as information in event detection and explanation of thermal comfort. However, correlations between continuous random variables are initial information based on physical perception. We also need to analyze in human levels if they are highly significant between the indoor environment, outdoor environment, and HVAC system. In the next section, we design measurement of discrete variables, whether they have statistical significance or not.

### 6.4. Correlations between Discrete RVs

This section aims to answer the research question “*Is there a statistical association between indoor environments, airflow patterns, and power consumption?*” that is based on observations from power consumption (**Ampere**). We begin our measurement using analysis of variance (ANOVA) to test the statistical significance of discrete random variables in the viewpoint of power consumption. Our goal is to indirectly measure airflow patterns of HVAC systems through power consumption of how they associate with indoor thermal comfort. Our case study is based on the reality that events randomly occur and are likely unbalanced. Moreover, we can understand that some RVs play an important role only when they interact, such as **TimeOfDay** and **Ampere**. It occurs independently on an everyday basis and has no statistical significance, but it becomes significantly important when conditioned with airflow behavior. Therefore, we measured our discrete RVs from Table 2 by choosing two-way ANOVA using the Type III method of sums of squares and setting the significance α level (alpha) at 0.05, as suggested by Larson [44]. The measurements using two-way ANOVA are shown in Table 7.

Table 7 summarizes the statistical significance of discrete RVs where **Target**, **TimeOfDay**, **In****-Out Temp Diff**, and **In_feellike** are perfectly relevant to **Ampere** (*p* value < 0.0001) except for **In****-Out Humi Diff** and **Thermostat Diff** that are lower than others (*p* values ~0.0478 and ~0.0278, respectively). Nevertheless, they still have statistical significance because *p* values are higher than the α level. When considering them together with **Target** based on two-way ANOVA (e.g., **X_1_** × **X_2_** means **X_1_** and **X_2_** co-influence **Y**), they become significantly important (*p* values < 0.0001). It confirms that our design of discrete RVs for thermal comfort measurement works and is ready to be applied in the event detection and explanation approach. However, we see that **Ex_feellike** was not included in the Table because the outdoor environment in our case study was too high humidity (~100% RH) and temperature (~34 °C) through the years. This situation is considered an unsuitable environment for humans to spend their time outside the building, and **Ex_feellike** becomes a constant with no significance with the others; thus, we excluded it from the Table.

We simplified this concern by visualizing a mathematical comparison of **Ampere** distributions when they are conditioned on a **TimeOfDay** in different **Target**. The mathematical comparisons are encoded in the form of a boxplot that is illustrated in Figure 5.

Figure 5 displays the overall distributions between good, bad, and worst conditions in the boxplot (boxplot of power off condition shows no information because its ampere factor has zero value). Boxplot represents the distributions of each event that the mid-point in the box (red lines) encodes for median values, and start and end points encode for minimum and maximum values. The area on the top median encodes 25% of its larger values, and the below-median in the box encodes 25% of its lower values.

We can see that these good and bad airflows are approximately equal (between four and seven), but they are entirely different when we consider the view of **TimeOfDay**. For example, the median of event distributions of good and bad airflows in the early morning is close, but areas on top and below red lines are different. It confirms that **Ampere** correlates to airflow conditions. In contrast, the event distributions conditioned on the worst airflow are different from the others. Their median and areas on top and below the red lines are shown in different shapes. We can say that the features of worst airflow are outstanding, and they are helpful to identify the events of indoor thermal comforts.

We can conclude that both continuous and discrete RVs are highly potential to be used in event detection and explanation of indoor thermal comfort. In the next section, we will approve our assumption by employing the RVs to train Deep Neural Networks that aim to model features for indoor thermal diagnosis.

## 7. Experimental Setup Based on Machine Learning

This section evaluates whether our predefined random variables, both continuous RVs and discrete RVs, can help software agents identify problems caused by the HVAC system semantically. RVs are modeled based on prior knowledge that represents human-like intelligence, particularly discrete RVs encoding for human-like interpretation. The research question is “*Do well-defined RVs from sensors help software agents predict HVAC system problems?*”.

### 7.1. Model Testing Metrics

We employed ML classification metrics that are precision, recall, and F-measure to evaluate model performance. Precision is a proportional ratio of correctly predicted outcomes against total predicted outcomes from models; recall is a proportional ratio of correctly predicted outcomes against actual outcomes from models; F-measure is a balanced point between precision and recall. We used the well-known python library—scikit-learn for the metrics computation.

The formula metrics of Precision, Recall, and F-measure are calculated as: $\mathrm{Precision}\;=\frac{\mathrm{TP}}{\mathrm{TP}+\mathrm{FP}}$, $\mathrm{Recall}\;=\frac{\mathrm{TP}}{\mathrm{TP}+\mathrm{FN}}$, $F-\mathrm{measure}\;=\frac{2\times \left(\mathrm{Precision}\;\times \;\mathrm{Recall}\right)}{\left(\mathrm{Precision}\;+\;\mathrm{Recall}\right)}$. True positives (TP) represent the case of corrected outcomes from models that are agreed with the actual predicted outcomes. False positives (FP) represent incorrect outcomes from models that disagree with the actual predicted outcomes. False negatives (FN) represent unidentified outcomes that are compared with the actual predicted outcomes.

According to the formulas, we can conclude that precision is concerned when the results of unidentified outcomes are detailed, and recall is considered when incorrect outcomes in the models are discussed. F-measure is considered when the medium between precision and recall is explained.

Because **Target** has multiple classes, we applied macro-averaged scores (Macro Avg) and weighted-average scores (Weighted Avg) to summarize the performance of each metric. The Macro Avg and Weighted Avg are calculated as follows.
(1)$$\mathrm{Macro}\;\mathrm{Avg}\;=\frac{\sum_{i=0}^{n}{\mathrm{metric}}_{i}}{n}.\;$$
(2)$$\mathrm{weighted}\mathrm{Avg}=\overset{n}{\underset{i=0}{\sum }}{\mathrm{weight}}_{i}\;\times \;{\mathrm{metric}}_{i}$$
(3)$${\mathrm{weight}}_{i}=\frac{{\mathrm{sample}}_{i}}{\sum_{i=0}^{n}s{\mathrm{ample}}_{i}}$$
where metrics are Precision, Recall, and F-measure, *n* is the number of events of **Target**, and ${\mathrm{sample}}_{i}$ is the number of event *i* samplings. The Macro Avg focused on all classes have the same priority without considering the number of classes in data samplings, which is needed to describe the overall model performance. The Weighted Avg focused on each class with a different priority according to the proportion of classes in data samplings. In simple words, minor classes lacking data samplings may cause poor performance, and Weighted Avg balances concern this by considering that each class with more proportion should be more significant.

### 7.2. Experimental Objectives

The experimental objective is to evaluate the effectiveness of ML models trained by RVs. The models’ predictive performance comparisons are proposed based on two scenarios: training based on continuous RVs representing unknown prior knowledge and fully learned from data. Training based on continuous RVs and discrete RVs represents learning from experts and observation. It conforms to how well-designed RVs can encourage software agents to learn and solve problems close to human-like intelligence.

The assumption is that well-design random variables will benefit them to learn and solve complex problems better. In simple words, our design and development of devices and RVs help ML algorithms approve learning ability to recognize the complex pattern from raw data.

### 7.3. Training and Testing Data Description

In total, 263,979 transactions were collected between May 16 June and 19 August 2021, in the environment set up in Section 6. Each transaction was transformed into 13 events as a state of random variables and their labeled classes detailed are in Section 4. However, the dataset was collected quickly with limited transactions, and partitioning training and testing data using 50–50 strategies may cause the model to be overfitting or underfitting. In this way, K-fold cross-validation was applied to slit testing and training data to avoid the over–under fitting problem. It is a concept to randomly divide training and testing data that employs all transactions to train and test the models.

K-fold cross-validation splits transactions into K sets where one of K sets is employed for testing data (e.g., unseen data in the model), and the rest, K − 1 folds, is employed for training sets. The configuration of this experiment sets k = 10, suggested by [45] where one holds for evaluating model’ performances, and nine utilizes for hyperparameter tuning. This solution aims to approve that the models can generalize problems and best-fit models when they need to deal with new data never seen before.

### 7.4. Results

The geometry, probability, and logical expression-based models were built using two hyperparameter tuning cases. First, partial hyperparameter tuning employs purely low-level information from sensors built without prior knowledge. Complete hyperparameter tuning utilizes low-level and high-level information, which means that the model was learned from raw data aligned with existing knowledge based on human-like understanding. They evaluated the performance using 10-fold cross-validation represented in Precision (PS), Recall (RC), and F-measure (F1). The results of partial hyperparameter tuning are shown in Table 8.

Table 8 illustrates that partial hyperparameter tuning-based models perform high performance for all approaches. The Weighted Avg and Weighted Avg show that both performances are equally likely. KNN can fit a model properly that can deal with unseen data with the highest performance. KNN works well with limited factors but has overwhelming training data to determine the distance between independent and dependent points. In contrast, NB is an underfitting model that produces the poorest RC in the worst airflow class. It means that NB cannot identify actual events such as Bad (52%) and Worst (44%) because NB assumes that all factors are independent, while Bad and Worst airflows are complex and dependent on factors to predict the output. The remaining models can perform acceptable performance.

We enhanced the assumption by supplementing prior knowledge based on high-level information to fine-tune hyperparameters. The results of complete hyperparameter tuning are shown in Table 9.

Table 9 shows that all models are improved compared with partial hyperparameter tuning-based models from Table 8. The outstanding model is DT in that Weighted Avg is up to 94% for all metrics. DT models high-level information well because it can compute categorical data, transform events using Boolean logic, and express most of our high-level information in categorical forms. Macro Avg of SVM, KNN, and ANN have perfect performance resulting around 90%. However, they are significantly different when considering Macro Avg F1. Their F1 Macro Avg is around 85%, slightly lower than DT. It can be summarized that geometry-based models perform less than logical expression-based models when most high-level information is represented in categorical forms. However, NB is slightly improved but has no statistical significance, although the high-level information is given. NB is unsuitable for complex problems, and providing new factors such as high-level information with complex conditions did not help the model turn hyperparameters well.

We simplified the results from complete and partial hyperparameter tuning models by comparing the overall performance based on Macro Avg, considering all classes are equally likely important and highlighting how models improve performance differently given additional prior knowledge. The comparative results are shown in Figure 6.

Figure 6 shows that the Precision and Recall of ANNs and DT improved around 10%, while SVM and KNN improved approximately around 5%. It means that given high-level information helps the model decrease incorrect outcomes (False Positive) and increases the effectiveness of dealing with unidentified outcomes (False Negative). However, BN is worse, given prior knowledge. It gave the model more incorrect outcomes because such prior knowledge was complex.

### 7.5. Discussion

In conclusion, our Internet of things-driven approach can help ML models solve thermal comfort problems caused by the HVAC system. The perception devices perform successfully with a valuable contribution of the open-source software and hardware. It is not possible to measure the real-time HVAC system data without technicians and engineers in general, but our approach could indirectly measure them based on prior knowledge. It imitates the way of technicians and engineers to detect and determine the HVAC system problems. We can say that well-design RVs help indirect models to identify fault events using stand sensors without breaking the HVAC system engine. The performance presents perfect accuracies, which is helpful for technicians and engineers to employ them and make better decisions. The logical expression-based model is suitable for thermal comfort problems when employing high-level information based on categorical forms. Because current prior knowledge encodes in discrete RVs, a logical expression-based model can compute ideally. It suggests that if prior knowledge is presented in continuous RVs with high dimensions, the geometry-based model might be a suitable solution to tune hyper parameters. In contrast, the probability-based model is not applicable for thermal comforts using complex factors.

This study has shown that machine learning models have been improved given prior knowledge. However, they need more contributions to improve performance. It is still challenging to fulfill research gaps; for example, the probability-based model is based on an advanced algorithm such as Causal Bayesian Networks, which is proposed to solve complex problems. Deep Neural Networks, an advanced algorithm of ANNs, are challenging to apply to thermal comfort fields to improve recent performance. Moreover, DT outperforms in our case study, but it is well known that DT is quite overfitting with training data. It can be applied to the Random Forests algorithm to advance the DT that deals with overfitting.

Although we mentioned that the indoor environment was fixed in the design and experiment section, more conditions are useful for the indoor thermal comfort analysis; for example, the variation of inside air velocity, which annotates the conduction effect of heat transfer and influences the thermal comfort conditions for the people in the room. Moreover, the number of people in the building space, which is interesting because the action of people in the room affects the air movement, complicates the heat flow dynamic in the room.

## 8. Conclusions

This study has addressed limitations in indoor thermal comfort that have helped engineers and technicians to monitor and detect problems on physical sites automatically. We proposed a new design and development of the Internet of Things-driven fault detection system to imitate human-like perception. It offers real-time monitoring of the problems caused by the HVAC system automatically. We represented prior knowledge based on expert experience and understanding to help the system detect the problem in the manner of human-like intelligence. Statistical significance between relevant factors from the Internet of Things device and prior knowledge is essential for monitoring and detecting HVAC system problems. We prove our proposed system using stand ML models based on geometry, probability, and logical expression for testing their predictive ability to identify HVAC system problems given sensor data and prior knowledge. The results show that models based on sensor data reached acceptable predictive performance, while additional prior knowledge could improve their predictive performance by around 10%. It reveals that the well design of prior knowledge can help systems identify problems in indoor thermal comfort. The sensor data based on the proposed device and prior knowledge are ready for real-world applications.

Although the recent study could monitor and detect problems caused by HVAC systems, it could not detail how and why such events happen. The current system cannot reason and explain the motivation behind the phenomenon, which is needed information when engineers and technicians have to make decisions. In the future, we will expand the ML model by augmenting cause–effect computing to help the model interpret event details close to human-like intelligence. Cause–effect computing is a new paradigm based on causal inference, such as attention mechanisms based on deep learning and structural causal mechanisms based on the directed acyclic graph. Applying them to the recent model that offers reasoning mechanisms and allows systems to detail the motivations behind problems is challenging.

Moreover, the recent case study considers HVAC system problems based on a single on–off engine in a small room. We will experiment on HVAC system inverter engines, multiple systems in a standard room such as a conference room and classroom. We will expand the period to 12–36 months to observe the hidden features in different seasons.

## Figures and Tables

**Figure 1 sensors-22-01925-f001:**
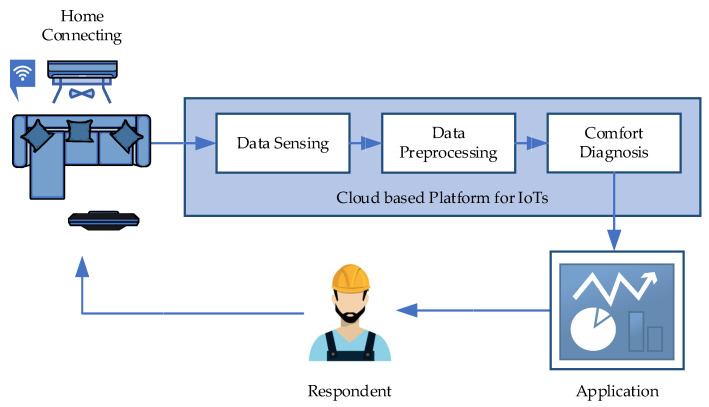
Overview architecture of fault detection and diagnosis of thermal comfort.

**Figure 2 sensors-22-01925-f002:**
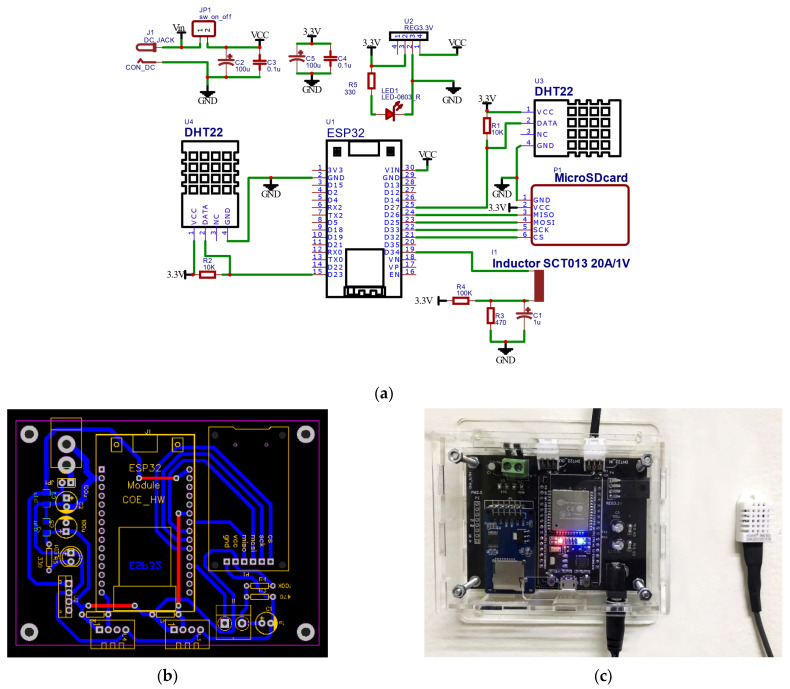
Schematic of the data acquisition setup as designed for this work: (**a**) diagram of electronic components and mechanical connectivity of device; (**b**) a printed circuit board (PCB) for the device electrical current flow; (**c**) prototype of sensor related devices.

**Figure 3 sensors-22-01925-f003:**
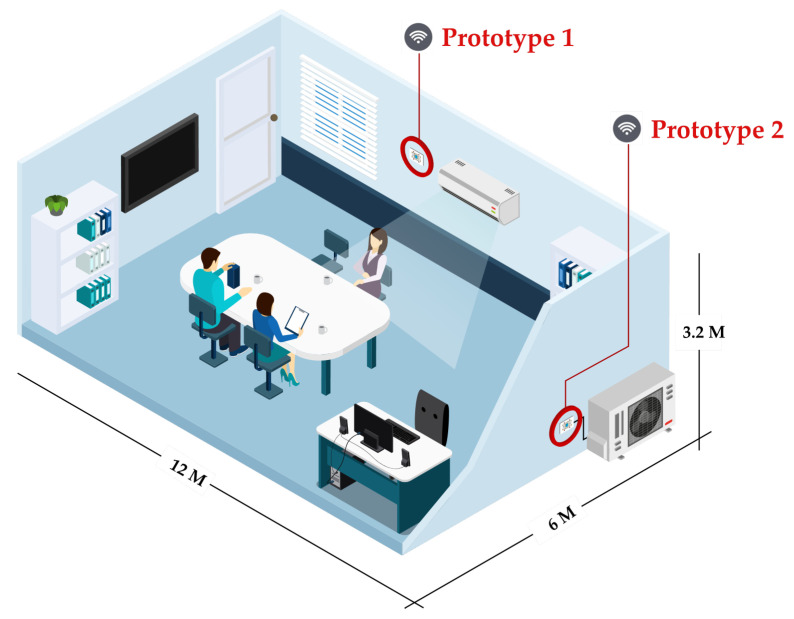
Installation of IoT-based measurement for thermal comfort in indoor environment.

**Figure 4 sensors-22-01925-f004:**
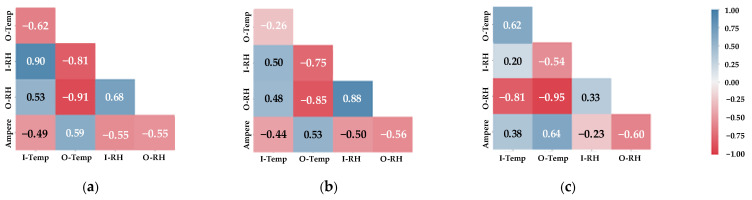
Thermal comfort correlations based on indoor environment, outdoor environment, and HVAC system: (**a**) good condition; (**b**) bad condition; (**c**) worst condition.

**Figure 5 sensors-22-01925-f005:**
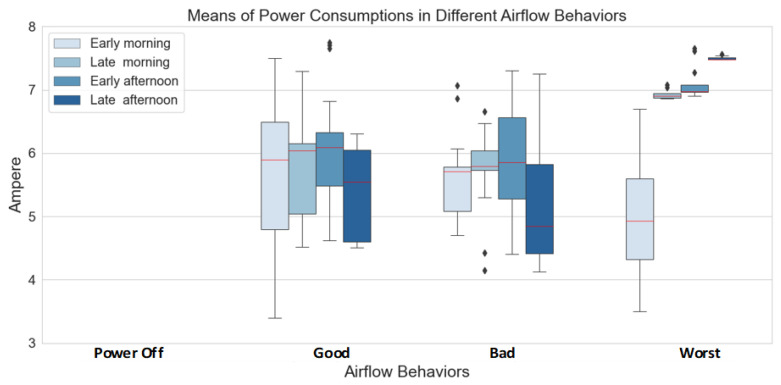
The differences of ampere means and errors conditioned on **TimeOfDay** and **Target**.

**Figure 6 sensors-22-01925-f006:**
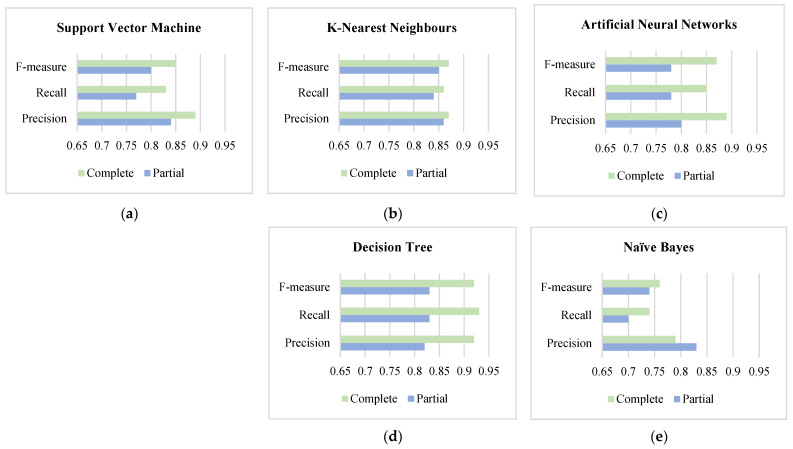
Comparison of the machine learning models between partial and complete hyperparameter tunings based on precision, recall, and f-measure: (**a**) comparison of the macro avg of SVM models; (**b**) comparison of the macro avg of KNN models; (**c**) comparison of the macro avg of ANN models; (**d**) comparison of the macro avg of DT models; (**e**) comparison of the macro avg of NB models.

**Table 1 sensors-22-01925-t001:** The low-level information based on RVs for thermal comfort measurement.

No.	Random Variable	State
1.	Indoor Temperature (I-Temp)	0.00–60.00 °C
2.	Indoor Relative Humidity (I-RH)	0–100% RH
3.	Outdoor Temperature (O-Temp)	0.00–60.00 °C
4.	Outdoor Humidity (O-RH)	0–100% RH
5.	Electric Current (Ampere)	0.00–20.00 A
6.	Timestamping (Date time)	24 h timestamp (YYYY-MM-DD HH:MM:SS)

**Table 2 sensors-22-01925-t002:** The high-level information based on RVs for thermal comfort measurement.

No.	Discrete RV	State	Primary Source
1.	HVAC Airflow	Good, Bad, Worst	Engineering Intervention
2.	Indoor Feeling	Comfortable, Dry, Uncomfortable, Worst	I-Temp and I-RH
3.	Outdoor Feeling	Comfortable, Dry, Uncomfortable, Worst	O-Temp and O-RH
4.	Time of Day	Morning, Afternoon, Evening, Night	Date Time
5.	In-Out Temp Diff	−20.00–20.00 °C	I-Temp and O-Temp
6.	In-Out Humi Diff	1.00–60.00% RH	I-RH and O-RH
7.	Thermostat Diff	1.00–10.00 °C	Thermostat and I-Temp

**Table 3 sensors-22-01925-t003:** The sensor models and their functions in interdisciplinary fields.

No.	Field	Physical Sensor Model	Factor	Range	Error Rate
1.	Indoor Environment	DHT22 AM2302	Temperature	−40–80 °C	±0.2 °C
			Relative Humidity	0–100% RH	±1.0% RH
2.	HVAC system	SCT-013	Ampere	0–100 A	≤0.2 A
3.	Outdoor Environment	AM2315 I2C Single Bus	Temperature	−40–125 °C	±0.3 °C
			Relative Humidity	0–100% RH	±2.0% RH

**Table 4 sensors-22-01925-t004:** Summary of good airflow condition.

Summary/Factor	I-Temp	I-RH	O-Temp	O-RH	Ampere
Minimum	24.80	41.20	30.10	70.20	0.14
1st Quartile	25.40	44.50	31.20	77.00	0.18
Median	25.50	46.40	31.80	79.20	6.94
3rd Quartile	25.70	47.80	32.10	81.80	7.71
Maximum	26.00	51.60	34.60	91.80	10.12
Mean	25.45	46.24	31.62	80.18	5.32
Standard Deviation	0.26	2.07	0.64	5.20	3.35

**Table 5 sensors-22-01925-t005:** Summary of bad airflow condition.

Summary/Factor	I-Temp	I-RH	O-Temp	O-RH	Ampere
Minimum	22.20	33.20	29.70	63.40	0.05
1st Quartile	22.60	37.10	30.90	71.40	6.47
Median	23.00	41.10	31.60	81.40	7.02
3rd Quartile	23.30	44.20	31.90	85.40	7.60
Maximum	23.90	54.00	35.00	96.20	9.93
**Mean**	**22** **.98**	**40** **.83**	**31** **.38**	**78** **.76**	**6** **.10**
**Standard Deviation**	**0** **.41**	**4** **.55**	**0** **.61**	**7** **.95**	**2** **.65**

**Table 6 sensors-22-01925-t006:** Summary of worst airflow condition.

Summary/Factor	I-Temp	I-RH	O-Temp	O-RH	Ampere
Minimum	23.00	31.10	27.20	65.30	0.01
1st Quartile	23.90	34.10	31.10	73.00	6.56
Median	26.50	40.10	31.80	76.40	7.10
3rd Quartile	28.40	40.60	32.30	83.70	7.61
Maximum	28.80	48.50	36.50	99.10	8.65
**Mean**	**26** **.15**	**38** **.09**	**31** **.35**	**79** **.17**	**6** **.85**
**Standard Deviation**	**2** **.08**	**3** **.66**	**1** **.51**	**9** **.18**	**1** **.54**

**Table 7 sensors-22-01925-t007:** The results of two-way ANOVA based on ampere-fixed observation.

Source	Degrees of Freedom	F Statistic	*p* Value
**Target**	3	2810.49	<0.0001
**In_feellike**	4	65.46	<0.0001 × 10^−55^
**TimeOfDay**	6	39.83	<0.0001 × 10^−48^
**In-Out Temp Diff**	1	1574.81	<0.0001
**In-Out Humi Diff**	1	54.79	<0.0001 × 10^−13^
**Thermostat Diff**	1	1094.35	<0.0001 × 10^−230^
**In****-Out Humi Diff** × **In_feellike**	4	62.69	<0.0001 × 10^−52^
Residual Variance	13177.0		

**Table 8 sensors-22-01925-t008:** The comparative effectiveness of **Target** based on partial hyperparameter tuning.

Model	SVM			KNN			ANN			DT			NB		
Class	PS	RC	F1	PS	RC	F1	PS	RC	F1	PS	RC	F1	PS	RC	F1
Power Off	0.96	0.92	0.94	0.96	0.94	0.95	0.95	0.91	0.93	0.95	0.95	0.95	0.97	0.92	0.94
Bad	0.77	0.63	0.69	0.76	0.73	0.75	0.64	0.75	0.69	0.72	0.70	0.71	0.74	0.52	0.61
Good	0.77	0.93	0.84	0.85	0.91	0.88	0.78	0.77	0.78	0.85	0.86	0.85	0.70	0.93	0.80
Worst	0.86	0.60	0.71	0.86	0.79	0.82	0.83	0.67	0.74	0.78	0.82	0.80	0.91	0.44	0.60
**Macro Avg**	**0** **.** **84**	**0** **.** **77**	**0** **.** **80**	**0** **.** **86**	**0** **.** **84**	**0** **.** **85**	**0** **.** **80**	**0** **.** **78**	**0** **.** **78**	**0** **.** **82**	**0** **.** **83**	**0** **.** **83**	**0** **.** **83**	**0** **.** **70**	**0** **.** **74**
**Weighted Avg**	**0** **.** **85**	**0** **.** **85**	**0** **.** **85**	**0** **.** **88**	**0** **.** **88**	**0** **.** **88**	**0** **.** **83**	**0** **.** **82**	**0** **.** **82**	**0** **.** **86**	**0** **.** **86**	**0** **.** **86**	**0** **.** **83**	**0** **.** **82**	**0** **.** **81**

**Table 9 sensors-22-01925-t009:** The comparative effectiveness of **Target** based on complete hyperparameter tuning.

Model	SVM			KNN			ANN			DT			NB		
Class	PS	RC	F1	PS	RC	F1	PS	RC	F1	PS	RC	F1	PS	RC	F1
Power Off	0.97	1.00	0.99	0.98	0.98	0.98	0.96	1.00	0.98	1.00	1.00	1.00	1.00	0.98	0.99
Bad	0.83	0.67	0.74	0.77	0.76	0.77	0.87	0.68	0.76	0.86	0.86	0.86	0.66	0.59	0.62
Good	0.85	0.95	0.90	0.87	0.90	0.88	0.86	0.94	0.90	0.92	0.91	0.92	0.77	0.88	0.82
Worst	0.89	0.69	0.78	0.86	0.80	0.83	0.89	0.76	0.82	0.90	0.94	0.92	0.73	0.52	0.61
**Macro Avg**	**0** **.** **89**	**0** **.** **83**	**0** **.** **85**	**0** **.** **87**	**0** **.** **86**	**0** **.** **87**	**0** **.** **89**	**0** **.** **85**	**0** **.** **87**	**0** **.** **92**	**0** **.** **93**	**0** **.** **92**	**0** **.** **79**	**0** **.** **74**	**0** **.** **76**
**Weighted Avg**	**0** **.** **90**	**0** **.** **90**	**0** **.** **90**	**0** **.** **90**	**0** **.** **90**	**0** **.** **90**	**0** **.** **90**	**0** **.** **91**	**0** **.** **90**	**0** **.** **94**	**0** **.** **94**	**0** **.** **94**	**0** **.** **84**	**0** **.** **84**	**0** **.** **84**

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
