# Peer review of "Design and Development of Internet of Things-Driven Fault Detection of Indoor Thermal Comfort: HVAC System Problems Case Study"

_sensors, 2022, doi:10.3390/s22051925_

Round 1

Reviewer 1 Report

I have reviewed the previous version and I am glad that authors have significantly improved the manuscript – it is now suitable for archival publication.

  1. Figure 2 has a prototype picture- is that yours? It has an overlay of “smart air care” on the image. Cannot tell if that is printed on the plastic casing or on the image. How about other figures? If you are copying them, all references should be provided in the figure caption and written permission should be obtained from the original publishers. Part a is too small to see anything. May be change the figure title to: “Figure 2. Schematic of the data acquisition setup as designed for this work”
  2. Figure 3 is a much better illustration than the previous version.
  3. Thanks for updating figure 4- it is now readable.
  4. RV is not a common acronym but it is used many times. If you can define it in a more prominent place than in the text- it will be easier to find. May be in the heading of section 4 in line 278. Random variables (RVs) for Environmental Representation.
  5. Last sentence of abstract does not make sense- what are “it”, “approves”, and “boot”? may be delete that sentence.
  6. Exemplified is not the correct use of the word in line 574. One usually exemplifies reward or punishment to set a lasting impression. Maybe you meant: “we used sample sizes of 1600 transactions to statistically analyze each airflow conditions.”

Author Response

Thank you for the opportunity to revise our manuscript, with an opportunity to address  your comments. We appreciate the time and effort that you have dedicated to providing your valuable feedback on our manuscript. We have been able to incorporate changes to reflect most of the suggestions provided by the reviewers.

Please see the attachment. We are uploading our point-by-point response to the comments (response to reviewers).

Best regards,
Sahoh, B., et at.

Reviewer 2 Report

My concerns are not addressed properly. 

Author Response

Thank you for the opportunity to revise our manuscript, with an opportunity to address your comments. We appreciate the time and effort that you have dedicated to providing your valuable feedback on our manuscript. 

We have carefully considered and we did our best in the last version, but still had not satisfied your concerns. Your comment is too short without a declaration which points are needed improvement. As we received conflicting advice from other reviewers that satisfies our revised manuscript, we decided to look forward to hearing from you in due time regarding our submission and to respond to any further questions and comments you may have.

Please see the attachment. We are uploading our point-by-point response to the comments (response to reviewers).

Best regards,
Sahoh, B., et at.

Reviewer 3 Report

In this paper, a IoT system to reduce false positives and false negatives in the indoor thermal comfort acknowledgement. I consider this paper interesting and useful to the scientific community.
Moreover, it is well-written and balance.

I only suggest to add more references concerning the existing literature in order to provide better performance comparison.

Concerning the existing systems and technologies, I suggest to add examples of advanced sensors for HVAC systems, e.g. radar-based detection:

E. Cardillo, C. Li, and A. Caddemi, “Embedded Heating, Ventilation, and Air Conditioning control systems: from traditional technologies towards radar advanced sensing” Review of Scientific Instruments, vol. 92, Issue 6, 061501, pp. 1-14, Jun. 2021.

Stephan M., Santra A., Fischer G. (2021) Human Target Detection and Localization with Radars Using Deep Learning. In: Wani M.A., Khoshgoftaar T.M., Palade V. (eds) Deep Learning Applications, Volume 2. Advances in Intelligent Systems and Computing, vol 1232. Springer, Singapore. https://doi.org/10.1007/978-981-15-6759-9_8.

Author Response

(The authors gave the same response as above.)

Reviewer 4 Report

The topic covered by the paper fits the scope of the Journal. After a careful revision, the following comments are provided for the enhancement of the manuscript.

In the keywords, “Internet of Things” could be added for coherence with the title.

The acronym IoT is directly used in the Abstract in line 23 but is should be introduced in previous line 21.

A similar issue occurs with MQTT, which appears in line 446 without indicating that it stands for Message Queue Telemetry Transport.

In line 59 “IoTs” appears and is not a term commonly used in scientific papers. If the authors refer to IoT devices or equipment, it should be expressed in such a way. The whole manuscript should be revised in this regard, including figures.

It is not required capitalizing all the words in figure captions or titles of tables.

In line 796 it seems that a citation is missing, only empty brackets are found.

Have the authors observed some malfunction or instability in the operation of the developed prototypes during continuous operation?

A version of Python, MicroPython is used in the proposal (subsection 5.2). It is a positive feature due to the fact that Python is a language widely used in research applications. The same comment is applicable to open source hardware like Esp32. In this regard, the trend of using open source software and hardware could be emphasized to justify the choice of such components, indeed with recent literature. Some papers that could be considered by the authors are now provided:

  • Open-Source MQTT-Based End-to-End IoT System for Smart City Scenarios. Future Internet 2022, 14, 57. https://doi.org/10.3390/fi14020057
  • Configurable IoT Open-Source Hardware and Software I-V Curve Tracer for Photovoltaic Generators. Sensors 2021, 21, 7650. https://doi.org/10.3390/s21227650
  • Elemental: An Open-Source Wireless Hardware and Software Platform for Building Energy and Indoor Environmental Monitoring and Control. Sensors 2019, 19, 4017. https://doi.org/10.3390/s19184017

The achieved results are well analyzed.

The fact of using open-source tools is a valuable contribution that could be highlighted in the subsection 7.5, Discussion. Additionally the main limitations of the work should be commented in a brief manner for a better presentation.

The references must be formatted according to the template of the Journal, namely, the abbreviated names of journals must be used, quotation marks are not required for article titles, etc.

Author Response

Thank you for the opportunity to revise our manuscript, with an opportunity to address your comments. We appreciate the time and effort that you have dedicated to providing your valuable feedback on our manuscript. We have been able to incorporate changes to reflect most of the suggestions provided by the reviewers.

Please see the attachment. We are uploading our point-by-point response to the comments (response to reviewers).

Best regards,
Sahoh, B., et at.

This manuscript is a resubmission of an earlier submission. The following is a list of the peer review reports and author responses from that submission.

Round 1

Reviewer 1 Report

The paper is very interesting. It is well written. The writers put a lot of effort into the paper explaining the general steps and the methods used.

Yet there are some details to be answered:

  1. Consideration of electric current supply is only for HVAC system or other system installed in the room as low-level information in thermal comfort measurement?
  2. What is the height of installation prototype 1 because there is change in thermal comfort with height variation?
  3. The paper does not discuss whether the results can be consistent with the variation of inside air velocity?
  4. Improve line 535; Sentence can’t start with a number.
  5. It will be helpful for the readers if a glimpse of weather data, the number of people considered in building space, and the pattern of occupancy are provided.

Reviewer 2 Report

Not a good presentation. Many items are missing. Trivial work with lots of effort. Need to add insight to make new contributions.

  1. Figure captions need to be improved. Like Figure 1 can be “Schematic data flow and analysis”. Should it not be a closed loop control diagram? Put more thought into this.
  2. What does figure 2 tell us? Did you develop this circuit or telling us how a circuit looks like? If you developed it – show an existing design without smart connections and then show your improvements and explain your contributions. Line 411 mentions Figure 2 shows the mechanical process of electrical design – but it does not show anything to the reader- please do a better job of illustration.
  3. Where are IoT s in Figure 3? It shows only the layouts and no sensors.
  4. Put the sample size of measurements in Table 4 and Table 5-6. How is a good and bad determined? Is it only you who decided on the comfort level or some external real people? If external real people- what was the sample size and distribution? Clarify bad and worst- why do you need both?
  5. Mention sample size for figure 4. Cannot read the numbers and the scale, Please increase font size.
  6. The summary of the work seems to be you collected data and then decided good, bad, and worst yourself. Then used classification techniques to identify the category. Not sure where the ML is applied? It is only classification with existing tools. Did you correct the settings based on the classification? What was the sample size and how did you verify?
  7. Figure 6 is output from some software. Identify the software and define the F-Measure, Recall, Precision. Different software and different versions can use different definitions.
  8. Does it have enough material to make an archival publication? Why should one read this and use this work? Try to emphasize your contributions and make the paper meaningful.

Reviewer 3 Report

The paper picked a very important and interesting topic, but the writing and presentation totally tarnished the potential merit embedded in the research. The paper is pretty lengthy, but fails to provide the most important information in the context. For the sake of clarity, the reviewer list the comments as bullet points: 

  1. the unique contribution of the research is not clearly stated and supported in the manuscript. The narratives about unique contributions are deviated from core argument by imply arguing thermal comfort and combining interdisciplinary factors are very important, but back up evidence. Your claim that “they lack integration of interdisciplinary research between HVAC systems, indoor and outdoor environments. This limitation becomes a problem in fault detection and diagnosis when engineers need to understand and deal with the real-world situations”, is too vague, which is unable to convince the readers that this is worth effort work.
  2. the claimed contributions in the manuscript seem not convincing, to the reviewer’s best knowledge, because there are already some publications regarding thermal comfort using machine learning and IOT technologies. The literature review section of this paper also admitted it, so how the authors allow such a conflict show up in the paper. The main contribution appears to be establishing the correlation between thermal comfort and interdisciplinary factors, which are not as claims all novel factors.
  3. since the main contribution of the research is adoption of IOT and machine learning in detecting malfunction of indoor thermal comfort, the paper should spend a lot of words descripting what the reasons are behind the system design and what is the specific critical configurations about the IOT system, what machine learning algorithm is used and the corresponding parameter are (e.g., layers and nodes, transformation functions, etc.). Nonetheless, these are all missing. The manuscript is full of non-essential statements.
  4. the writing is another significant weakness. The paper is extremely poorly structured because the theoretical aspect of the paper is not proper written and the case study took too much volume. Another serious problem is that the writing does not read like a scientific article because there lacks of reasoning about most arguments. The authors should be explaining or arguing why you pick this topic and why this ,why that. Instead, the entire manuscript is full of unsupported claims.